# Sparse Additive Text Models with Low Rank Background

**Lei Shi**
Baidu.com, Inc.
P.R. China
shilei06@baidu.om

## Abstract

The sparse additive model for text modeling involves the sum-of-exp computing, whose cost is consuming for large scales. Moreover, the assumption of equal background across all classes/topics may be too strong. This paper extends to propose sparse additive model with low rank background (SAM-LRB) and obtains simple yet efficient estimation. Particularly, employing a double majorization bound, we approximate log-likelihood into a quadratic lower-bound without the log-sum-exp terms. The constraints of low rank and sparsity are then simply embodied by nuclear norm and $\ell_1$-norm regularizers. Interestingly, we find that the optimization task of SAM-LRB can be transformed into the same form as in Robust PCA. Consequently, parameters of supervised SAM-LRB can be efficiently learned using an existing algorithm for Robust PCA based on accelerated proximal gradient. Besides the supervised case, we extend SAM-LRB to favor unsupervised and multifaceted scenarios. Experiments on three real data demonstrate the effectiveness and efficiency of SAM-LRB, compared with a few state-of-the-art models.

## 1 Introduction

Generative models of text have gained large popularity in analyzing a large collection of documents [3, 4, 17]. This type of models overwhelmingly rely on the Dirichlet-Multinomial conjugate pair, perhaps mainly because its formulation and estimation is straightforward and efficient. However, the ease of parameter estimation may come at a cost: unnecessarily over-complicated latent structures and lack of robustness to limited training data. Several efforts emerged to seek alternative formulations, taking the correlated topic models [13, 19] for instance.

Recently in [10], the authors listed three main problems with Dirichlet-Multinomial generative models, namely inference cost, overparameterization, and lack of sparsity. Motivated by them, a Sparse Additive GEnerative model (SAGE) was proposed in [10] as an alternative choice of generative model. Its core idea is that the lexical distribution in log-space comes by adding the background distribution with sparse deviation vectors. Successfully applying SAGE, effort [14] discovers geographical topics in the twitter stream, and paper [25] detects communities in computational linguistics.

However, SAGE still suffers from two problems. First, the likelihood and estimation involve the sum-of-exponential computing due to the soft-max generative nature, and it would be time consuming for large scales. Second, SAGE assumes one single background vector across all classes/topics, or equivalently, there is one background vector for each class/topic but all background vectors are constrained to be equal. This assumption might be too strong in some applications, e.g., when lots of synonyms vary their distributions across different classes/topics.

Motivated to solve the second problem, we are propose to use a low rank constrained background. However, directly assigning the low rank assumption to the log-space is difficult. We turn to approximate the data log-likelihood of sparse additive model by a quadratic lower-bound based on the

double majorization bound in [6], so that the costly log-sum-exponential computation, i.e., the first problem of SAGE, is avoided. We then formulate and derive learning algorithm to the proposed SAM-LRB model. Main contributions of this paper can be summarized into four-fold as below:

- Propose to use low rank background to extend the equally constrained setting in SAGE.
- Approximate the data log-likelihood of sparse additive model by a quadratic lower-bound based on the double majorization bound in [6], so that the costly log-sum-exponential computation is avoided.
- Formulate the constrained optimization problem into Lagrangian relaxations, leading to a form exactly the same as in Robust PCA [28]. Consequently, SAM-LRB can be efficiently learned by employing the accelerated proximal gradient algorithm for Robust PCA [20].
- Extend SAM-LRB to favor supervised classification, unsupervised topic model and multi-faceted model; conduct experimental comparisons on real data to validate SAM-LRB.

## 2 Supervised Sparse Additive Model with Low Rank Background

### 2.1 Supervised Sparse Additive Model

Same as in SAGE [10], the core idea of our model is that the lexical distribution in log-space comes from adding the background distribution with additional vectors. Particularly, we are given documents $D$ documents over $M$ words. For each document $d \in [1, D]$, let $y_d \in [1, K]$ represent the class label in the current supervised scenario, $\boldsymbol{c}_d \in \mathbb{R}_+^M$ denote the vector of term counts, and $C_d = \sum_w c_{dw}$ be the total term count. We assume each class $k \in [1, K]$ has two vectors $\boldsymbol{b}_k, \boldsymbol{s}_k \in \mathbb{R}^M$, denoting the background and additive distributions in log-space, respectively. Then the generative distribution for each word $w$ in a document $d$ with label $y_d$ is a soft-max form:

$$p(w|y_d) = p(w|y_d, \boldsymbol{b}_{y_d}, \boldsymbol{s}_{y_d}) = \frac{\exp(b_{y_d w} + s_{y_d w})}{\sum_{i=1}^M \exp(b_{y_d i} + s_{y_d i})}. \tag{1}$$

Given $\boldsymbol{\Theta} = \{\boldsymbol{B}, \boldsymbol{S}\}$ with $\boldsymbol{B} = [\boldsymbol{b}_1, \ldots, \boldsymbol{b}_K]$ and $\boldsymbol{S} = [\boldsymbol{s}_1, \ldots, \boldsymbol{s}_K]$, the log-likelihood of data $\mathcal{X}$ is:

$$\mathcal{L} = \log p(\mathcal{X}|\boldsymbol{\Theta}) = \sum_{k=1}^K \sum_{d:y_d=k} \mathcal{L}(d, k), \quad \mathcal{L}(d, k) = \boldsymbol{c}_d^\top (\boldsymbol{b}_k + \boldsymbol{s}_k) - C_d \log \sum_{i=1}^M \exp(b_{ki} + s_{ki}). \tag{2}$$

Similarly, a testing document $d$ is classified into class $\hat{y}(d)$ according to $\hat{y}(d) = \arg\max_k \mathcal{L}(d, k)$. In SAGE [10], the authors further assumed that the background vectors across all classes are the same, i.e., $\boldsymbol{b}_k = \boldsymbol{b}$ for $\forall k$, and each additive vector $\boldsymbol{s}_k$ is sparse. Although intuitive, the background equality assumption may be too strong for real applications. For instance, to express a same/similar meaning, different classes of documents may choose to use different terms from a tuple of synonyms. In this case, SAGE would tend to include these terms as the sparse additive part, instead of as the background. Taking Fig. 1 as an illustrative example, the log-space distribution (left) is the sum of the low-rank background $\boldsymbol{B}$ (middle) and the sparse $\boldsymbol{S}$ (right). Applying SAGE to this type of data, the equality constrained background $\boldsymbol{B}$ would fail to capture the low-rank structure, and/or the additive part $\boldsymbol{S}$ would be not sparse, so that there may be risks of over-fitting or under-fitting.

Moreover, since there exists sum-of-exponential terms in Eq. (2) and thus also in its derivatives, the computing cost becomes huge when the vocabulary size $M$ is large. As a result, although performing well in [10, 14, 25], SAGE might still suffer from problems of over-constrain and inefficiency.

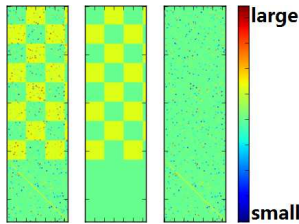

Figure 1: Low rank background. Left to right illustrates the log-space distr., background $\boldsymbol{B}$, and sparse $\boldsymbol{S}$, resp. Rows index terms, and columns for classes.

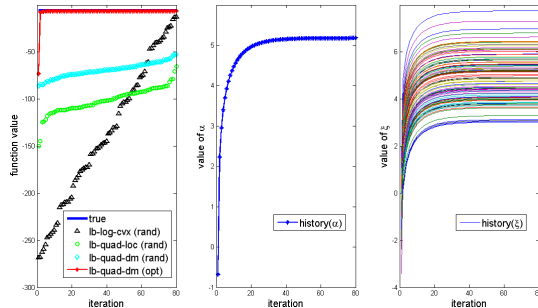

Figure 2: Lower-bound's optimization. Left to right shows the trajectory of lower-bound, $\alpha$, and $\boldsymbol{\xi}$, resp.

## 2.2 Supervised Sparse Additive Model with Low Rank Background

Motivated to avoid the inefficient computing due to sum-of-exp, we adopt the double majorization lower-bound of $\mathcal{L}$ [6], so that it is well approximated and quadratic w.r.t. $\boldsymbol{B}$ and $\boldsymbol{S}$. Further based on this lower-bound, we proceed to assume the background $\boldsymbol{B}$ across classes is low-rank, in contrast to the equality constraint in SAGE. An optimization algorithm is proposed based on proximal gradient.

### 2.2.1 Double Majorization Quadratic Lower Bound

In the literature, there have been several existing efforts on efficient computing the sum-of-exp term involved in soft-max [5, 15, 6]. For instance, based on the convexity of logarithm, one can obtain a bound $-\log \sum_i \exp(x_i) \geq -\phi \sum_i \exp(x_i) + \log \phi + 1$ for any $\phi \in \mathbb{R}_+$, namely the `lb-log-cvx` bound. Moreover, via upper-bounding the Hessian matrix, one can obtain the following local quadratic approximation for any $\forall \xi_i \in \mathbb{R}$, shortly named as `lb-quad-loc`:

$$-\log \sum_{i=1}^M \exp(x_i) \geq \frac{1}{M}(\sum_i x_i - \sum_i \xi_i)^2 - \sum_i (x_i - \xi_i)^2 - \frac{\sum_i (x_i - \xi_i)\exp(\xi_i)}{\sum_i \exp(\xi_i)} - \log \sum_i \exp(\xi_i).$$

In [6], Bouchard proposed the following quadratic lower-bound by double majorization (`lb-quad-dm`) and demonstrated its better approximation compared with the previous two:

$$-\log \sum_{i=1}^M \exp(x_i) \geq -\alpha - \frac{1}{2} \sum_{i=1}^M \left\{ x_i - \alpha - \xi_i + f(\xi_i)[(x_i - \alpha)^2 - \xi_i^2] + 2\log[\exp(\xi_i) + 1] \right\}, \quad (3)$$

with $\alpha \in \mathbb{R}$ and $\boldsymbol{\xi} \in \mathbb{R}_+^M$ being auxiliary (variational) variables, and $f(\xi) = \frac{1}{2\xi} \cdot \frac{\exp(\xi)-1}{\exp(\xi)+1}$. This bound is closely related to the bound proposed by Jaakkola and Jordan [6].

Employing Eq. (3), we obtain a lower-bound $\mathcal{L}_{lb} \leq \mathcal{L}$ to the data log-likelihood in Eq. (2):

$$\mathcal{L}_{lb} = \sum_{k=1}^K \left[ -(\boldsymbol{b}_k + \boldsymbol{s}_k)^\top \boldsymbol{A}_k (\boldsymbol{b}_k + \boldsymbol{s}_k) - \boldsymbol{\beta}_k^\top (\boldsymbol{b}_k + \boldsymbol{s}_k) - \gamma_k \right],$$

$$\text{with } \gamma_k = \tilde{C}_k \left\{ \alpha_k - \frac{1}{2} \sum_{i=1}^M \left[ \alpha_k + \xi_{ki} + f(\xi_{ki})(\alpha_k^2 - \xi_{ki}^2) + 2\log(\exp(\xi_{ki}) + 1) \right] \right\},$$

$$\boldsymbol{A}_k = \tilde{C}_k \text{diag}\left[ f(\boldsymbol{\xi}_k) \right], \quad \boldsymbol{\beta}_k = \tilde{C}_k (\frac{1}{2} - \alpha_k f(\boldsymbol{\xi}_k)) - \sum_{d:y_d=k} \boldsymbol{c}_d, \quad \tilde{C}_k = \sum_{d:y_d=k} C_d. \quad (4)$$

For each class $k$, the two variational variables, $\alpha_k \in \mathbb{R}$ and $\boldsymbol{\xi}_k \in \mathbb{R}_+^M$, can be updated iteratively as below for a better approximated lower-bound. Therein, $\text{abs}(\cdot)$ denotes the absolute value operator.

$$\alpha_k = \frac{1}{\sum_{i=1}^M f(\xi_{ki})} \left[ \frac{M}{2} - 1 + \sum_{i=1}^M (b_{ki} + s_{ki}) f(\xi_{ki}) \right], \qquad \boldsymbol{\xi}_k = \text{abs}(\boldsymbol{b}_k + \boldsymbol{s}_k - \alpha_k). \quad (5)$$

One example of the trajectories during optimizing this lower-bound is illustrated in Fig. 2. Particularly, the left shows the lower-bound converges quickly to ground truth, usually within 5 rounds in our experiences. The values of the three lower-bounds with randomly sampled the variational variables are also sorted and plotted. One can find that `lb-quad-dm` approximates better or comparably well even with a random initialization. Please see [6] for more comparisons.

### 2.2.2 Supervised SAM-LRB Model and Optimization by Proximal Gradient

Rather than optimizing the data log-likelihood in Eq. (2) like in SAGE, we turn to optimize its lower-bound in Eq. (4), which is convenient for further assigning the low-rank constraint on $\boldsymbol{B}$ and the sparsity constraint on $\boldsymbol{S}$. Concretely, our target is formulated as a constrained optimization task:

$$\max_{\boldsymbol{B}, \boldsymbol{S}} \quad \mathcal{L}_{lb}, \qquad \text{with } \mathcal{L}_{lb} \text{ specified in Eq. (4)},$$

$$\text{s.t.} \quad \boldsymbol{B} = [\boldsymbol{b}_1, \ldots, \boldsymbol{b}_K] \text{ is low rank}, \qquad \boldsymbol{S} = [\boldsymbol{s}_1, \ldots, \boldsymbol{s}_K] \text{ is sparse}. \quad (6)$$

Concerning the two constraints, we call the above as supervised **S**parse **A**dditive **M**odel with **Low**-**R**ank **B**ackground, or supervised SAM-LRB for short. Although both of the two assumptions can

be tackled via formulating a fully generative model, assigning appropriate priors, and delivering inference in a Bayesian manner similar to [8], we determine to choose the constrained optimization form for not only a clearer expression but also a simpler and efficient algorithm.

In the literature, there have been several efforts considering both low rank and sparse constraints similar to Eq. (6), most of which take the use of proximal gradient [2, 7]. Papers [20, 28] studied the problems under the name of Robust Principal Component Analysis (RPCA), aiming to decouple an observed matrix as the sum of a low rank matrix and a sparse matrix. Closely related to RPCA, our scenario in Eq. (6) can be regarded as a weighted RPCA formulation, and the weights are controlled by variational variables. In [24], the authors proposed an efficient algorithm for problems that constrain a matrix to be both low rank and sparse simultaneously.

Following these existing works, we adopt the nuclear norm to implement the low rank constraint, and $\ell_1$-norm for the sparsity constraint, respectively. Letting the partial derivative w.r.t. $\boldsymbol{\lambda}_k = (\boldsymbol{b}_k + \boldsymbol{s}_k)$ of $\mathcal{L}_{lb}$ equal to zero, the maximum of $\mathcal{L}_{lb}$ can be achieved at $\boldsymbol{\lambda}_k^* = -\frac{1}{2}\boldsymbol{A}_k^{-1}\boldsymbol{\beta}_k$. Since $\boldsymbol{A}_k$ is positive definite and diagonal, the optimal solution $\boldsymbol{\lambda}_k^*$ is well-posed and can be efficiently computed. Simultaneously considering the equality $\boldsymbol{\lambda}_k = (\boldsymbol{b}_k + \boldsymbol{s}_k)$, the low rank on $\boldsymbol{B}$ and the sparsity on $\boldsymbol{S}$, one can rewritten Eq. (6) into the following Lagrangian form:

$$\min_{\boldsymbol{B},\boldsymbol{S}} \quad \frac{1}{2}||\boldsymbol{\Lambda}^* - \boldsymbol{B} - \boldsymbol{S}||_F^2 + \mu(||\boldsymbol{B}||_* + \nu|\boldsymbol{S}|_1), \quad \text{with } \boldsymbol{\Lambda}^* = [\boldsymbol{\lambda}_1^*, \ldots, \boldsymbol{\lambda}_K^*], \qquad (7)$$

where $||\cdot||_F$, $||\cdot||_*$ and $|\cdot|_1$ denote the Frobenius norm, nuclear norm and $\ell_1$-norm, respectively. The Frobenius norm term concerns the accuracy of decoupling from $\boldsymbol{\Lambda}^*$ into $\boldsymbol{B}$ and $\boldsymbol{S}$. Lagrange multipliers $\mu$ and $\nu$ control the strengths of low rank constraint and sparsity constraint, respectively.

Interestingly, Eq. (7) is exactly the same as the objective of RPCA [20, 28]. Paper [20] proposed an algorithm for RPCA based on *accelerated* proximal gradient (APG-RPCA), showing its advantages of efficiency and stability over (plain) proximal gradient. We choose it, i.e., Algorithm 2 in [20], for seeking solutions to Eq. (7). The computations involved in APG-RPCA include SVD decomposition and absolute value thresholding, and interested readers are referred to [20] for more details. The augmented Lagrangian and alternating direction methods [9, 29] could be considered as alternatives.

---

**Data**: Term counts and labels $\{\boldsymbol{c}_d, C_d, y_d\}_{d=1}^D$ of $D$ docs and $K$ classes, sparse thres. $\nu \approx 0.05$
**Result**: Log-space distributions: low-rank $\boldsymbol{B}$ and sparse $\boldsymbol{S}$
**Initialization**: randomly initialize parameters $\{\boldsymbol{B}, \boldsymbol{S}\}$, and variational variables $\{\alpha_k, \boldsymbol{\xi}_k\}_k$;
**while** *not converge* **do**
    **if** *optimize variational variables* **then** iteratively update $\{\alpha_k, \boldsymbol{\xi}_k\}_k$ according to Eq. (5);
    **for** $k = 1, \ldots, K$ **do** calculate $\boldsymbol{A}_k$ and $\boldsymbol{\beta}_k$ by Eq. (4), and $\boldsymbol{\lambda}_k^* = -\frac{1}{2}\boldsymbol{A}_k^{-1}\boldsymbol{\beta}_k$ ;
    $\boldsymbol{B}, \boldsymbol{S} \longleftarrow$ APG-RPCA$(\boldsymbol{\Lambda}^*, \nu)$ by Algorithm 2 in [20], with $\boldsymbol{\Lambda}^* = [\boldsymbol{\lambda}_1^*, \ldots, \boldsymbol{\lambda}_K^*]$;
**end**

**Algorithm 1:** Supervised SAM-LRB learning algorithm

---

Consequently, the supervised SAM-LRB algorithm is specified in Algorithm 1. Therein, one can choose to either fix or update the variational variables $\{\alpha_k, \boldsymbol{\xi}_k\}_k$. If they are fixed, Algorithm 1 has only one outer iteration with no need to check the convergence. Compared with the supervised SAGE learning algorithm in Sec. 3 of [10], our supervised SAM-LRB algorithm not only does not need to compute the sum of exponentials so that computing cost is saved, but also is optimized simply and efficiently by proximal gradient instead of using Newton updating as in SAGE. Moreover, adding Laplacian-Exponential prior on $\boldsymbol{S}$ for sparseness, SAGE updates the conjugate posteriors and needs to employ a "warm start" technique to avoid being trapped in early stages with inappropriate initializations, while in contrast SAM-LRB does not have this risk. Additionally, since the evolution from SAGE to SAM-LRB is two folded, i.e., the low rank background assumption and the convex relaxation, we find that adopting the convex relaxation also helps SAGE during optimization.

## 3  Extensions

Analogous to [10], our SAM-LRB formulation can be also extended to unsupervised topic modeling scenario with latent variables, and the scenario with multifaceted class labels.

## 3.1 Extension 1: Unsupervised Latent Variable Model

We consider how to incorporate SAM-LRB in a latent variable model of unsupervised text modelling. Following topic models, there is one latent vector of topic proportions per document and one latent discrete variable per term. That is, each document $d$ is endowed with a vector of topic proportions $\boldsymbol{\theta}_d \sim \mathrm{Dirichlet}(\rho)$, and each term $w$ in this document is associated with a latent topic label $z_w^{(d)} \sim \mathrm{Multinomial}(\boldsymbol{\theta}_d)$. Then the probability distribution for $w$ is

$$p(w|z_w^{(d)}, \boldsymbol{B}, \boldsymbol{S}) \propto \exp\left( b_{z_w^{(d)}w} + s_{z_w^{(d)}w} \right), \tag{8}$$

which only replaces the known class label $y_d$ in Eq. (1) with the unknown topic label $z_w^{(d)}$.

We can combine the mean field variational inference for latent Dirichlet allocation (LDA) [4] with the lower-bound treatment in Eq. (4), leading to the following unsupervised lower-bound

$$
\begin{aligned}
\mathcal{L}_{lb} = & \sum_{k=1}^{K} \left[ -(\boldsymbol{b}_k + \boldsymbol{s}_k)^\top \boldsymbol{A}_k (\boldsymbol{b}_k + \boldsymbol{s}_k) - \boldsymbol{\beta}_k^\top (\boldsymbol{b}_k + \boldsymbol{s}_k) - \gamma_k \right] \\
& + \sum_d \left[ \langle \log p(\boldsymbol{\theta}_d|\rho) \rangle - \langle \log Q(\boldsymbol{\theta}_d) \rangle \right] + \sum_d \sum_w \left[ \langle \log p(z_w^{(d)}|\boldsymbol{\theta}_d) \rangle - \langle \log Q(z_w^{(d)}) \rangle \right],
\end{aligned}
$$

$$\text{with } \gamma_k = \tilde{C}_k \left\{ \alpha_k - \frac{1}{2} \sum_{i=1}^{M} \left[ \alpha_k + \xi_{ki} + f(\xi_{ki})(\alpha_k^2 - \xi_{ki}^2) + 2\log(\exp(\xi_{ki}) + 1) \right] \right\},$$

$$\boldsymbol{A}_k = \tilde{C}_k \mathrm{diag}\left[ f(\boldsymbol{\xi}_k) \right], \qquad \boldsymbol{\beta}_k = \tilde{C}_k (\frac{1}{2} - \alpha_k f(\boldsymbol{\xi}_k)) - \tilde{\boldsymbol{c}}_k, \tag{9}$$

where each $w$-th item in $\tilde{\boldsymbol{c}}_k$ is $\tilde{c}_{kw} = \sum_d Q(k|d, w)c_{dw}$, i.e. the expected count of term $w$ in topic $k$, and $\tilde{C}_k = \sum_w \tilde{c}_{kw}$ is the topic's expected total count throughout all words.

This unsupervised SAM-LRB model formulates a topic model with low rank background and sparse deviation, which is learned via EM iterations. The E-step to update posteriors $Q(\boldsymbol{\theta}_d)$ and $Q(z_w^{(d)})$ is identical to the standard LDA. Once $\{\boldsymbol{A}_k, \boldsymbol{\beta}_k\}$ are computed as above, the M-step to update $\{\boldsymbol{B}, \boldsymbol{S}\}$ and variational variables $\{\alpha_k, \boldsymbol{\xi}_k\}_k$ remains the same as the supervised case in Algorithm 1.

## 3.2 Extension 2: Multifaceted Modelling

We consider how SAM-LRB can be used to combine multiple facets (multi-dimensional class labels), i.e, combining per-word latent topics and document labels and pursuing a structural view of labels and topics. In the literature, multifaceted generative models have been studied in [1, 21, 23], and they incorporated latent switching variables that determine whether each term is generated from a topic or from a document label. Topic-label interactions can also be included to capture the distributions of words at the intersections. However in this kind of models, the number of parameters becomes very large for large vocabulary size, many topics, many labels. In [10], SAGE needs no switching variables and shows advantageous of model sparsity on multifaceted modeling. More recently, paper [14] employs SAGE and discovers meaningful geographical topics in the twitter streams.

Applying SAM-LRB to the multifaceted scenario, we still assume the multifaceted variations are composed of low rank background and sparse deviation. Particularly, for each topic $k \in [1, K]$, we have the topic background $\boldsymbol{b}_k^{(T)}$ and sparse deviation $\boldsymbol{s}_k^{(T)}$; for each label $j \in [1, J]$, we have label background $\boldsymbol{b}_j^{(L)}$ and sparse deviation $\boldsymbol{s}_j^{(L)}$; for each topic-label interaction pair $(k, j)$, we have only the sparse deviation $\boldsymbol{s}_{kj}^{(I)}$. Again, background distributions $\boldsymbol{B}^{(T)} = [\boldsymbol{b}_1^{(T)}, \ldots, \boldsymbol{b}_K^{(T)}]$ and $\boldsymbol{B}^{(L)} = [\boldsymbol{b}_1^{(L)}, \ldots, \boldsymbol{b}_J^{(L)}]$ are assumed of low ranks to capture single view's distribution similarity.

Then for a single term $w$ given the latent topic $z_w^{(d)}$ and the class label $y_d$, its generative probability is obtained by summing the background and sparse components together:

$$p(w|z_w^{(d)}, y_d, \boldsymbol{\Theta}) \propto \exp\left( b_{z_w^{(d)}w}^{(T)} + s_{z_w^{(d)}w}^{(T)} + b_{y_dw}^{(L)} + s_{y_dw}^{(L)} + s_{z_w^{(d)}y_dw}^{(I)} \right), \tag{10}$$

with parameters $\boldsymbol{\Theta} = \{\boldsymbol{B}^{(T)}, \boldsymbol{S}^{(T)}, \boldsymbol{B}^{(L)}, \boldsymbol{S}^{(L)}, \boldsymbol{S}^{(I)}\}$. The log-likelihood's lower-bound involves the sum through all topic-label pairs:

$$
\begin{aligned}
\mathcal{L}_{lb} \;=\; & \sum_{k=1}^{K} \sum_{j=1}^{J} \left[ -\boldsymbol{\lambda}_{kj}^{\top} \boldsymbol{A}_{kj} \boldsymbol{\lambda}_{kj} - \boldsymbol{\beta}_{kj}^{\top} \boldsymbol{\lambda}_{kj} - \gamma_{kj} \right] \\
& + \sum_{d} \left[ \langle \log p(\boldsymbol{\theta}_d | \rho) \rangle - \langle \log Q(\boldsymbol{\theta}_d) \rangle \right] + \sum_{d} \sum_{w} \left[ \langle \log p(z_w^{(d)} | \boldsymbol{\theta}_d) \rangle - \langle \log Q(z_w^{(d)}) \rangle \right],
\end{aligned}
$$
with
$$
\boldsymbol{\lambda}_{kj} \triangleq \boldsymbol{b}_k^{(T)} + \boldsymbol{s}_k^{(T)} + \boldsymbol{b}_j^{(L)} + \boldsymbol{s}_j^{(L)} + \boldsymbol{s}_{kj}^{(I)}. \tag{11}
$$

In the quadratic form, the values of $\boldsymbol{A}_{kj}$, $\boldsymbol{\beta}_{kj}$ and $\gamma_{kj}$ are trivial combination of Eq. (4) and Eq. (9), i.e., weighted by both the observed labels and posteriors of latent topics. Details are omitted here due to space limit. The second row remains the same as in Eq. (9) and standard LDA.

During the iterative estimation, every iteration includes the following steps:

- Estimate the posteriors $Q(z_w^{(d)})$ and $Q(\boldsymbol{\theta}_d)$;
- With $(\boldsymbol{B}^{(T)}, \boldsymbol{S}^{(T)}, \boldsymbol{S}^{(I)})$ fixed, solve a quadratic program over $\boldsymbol{\Lambda}^{*(L)}$, which approximates the sum of $\boldsymbol{B}^{(L)}$ and $\boldsymbol{S}^{(L)}$. Put $\boldsymbol{\Lambda}^{*(L)}$ into Algorithm 1 to update $\boldsymbol{B}^{(L)}$ and $\boldsymbol{S}^{(L)}$;
- With $(\boldsymbol{B}^{(L)}, \boldsymbol{S}^{(L)}, \boldsymbol{S}^{(I)})$ fixed, solve a quadratic program over $\boldsymbol{\Lambda}^{*(T)}$, which approximates the sum of $\boldsymbol{B}^{(T)}$ and $\boldsymbol{S}^{(T)}$. Put $\boldsymbol{\Lambda}^{*(T)}$ into Algorithm 1 to update $\boldsymbol{B}^{(T)}$ and $\boldsymbol{S}^{(T)}$;
- With $(\boldsymbol{B}^{(T)}, \boldsymbol{S}^{(T)}, \boldsymbol{B}^{(L)}, \boldsymbol{S}^{(L)})$ fixed, update $\boldsymbol{S}^{(I)}$ by proximal gradient.

## 4 Experimental Results

In order to test SAM-LRB in different scenarios, this section considers experiments under three tasks, namely supervised document classification, unsupervised topic modeling, and multi-faceted modeling and classification, respectively.

### 4.1 Document Classification

We first test our SAM-LRB model in the supervised document modeling scenario and evaluate the classification accuracy. Particularly, the supervised SAM-LRB is compared with the Dirichlet-Multinomial model and SAGE. The precision of the Dirichlet prior in Dirichlet-Multinomial model is updated by the Newton optimization [22]. Nonparametric Jeffreys prior [12] is adopted in SAGE as a parameter-free sparse prior. Concerning the variational variables $\{\alpha_i, \boldsymbol{\xi}_i\}_i$ in the quadratic lower-bound of SAM-LRB, both cases of fixing them and updating them are considered.

We consider the benchmark `20Newsgroups` data[1], and aim to classify unlabelled newsgroup postings into 20 newsgroups. No stopword filtering is performed, and we randomly pick a vocabulary of 55,000 terms. In order to test the robustness, we vary the proportion of training data. After 5 independent runs by each algorithm, the classification accuracies on testing data are plotted in Fig. 3 in terms of box-plots, where the lateral axis varies the training data proportion.

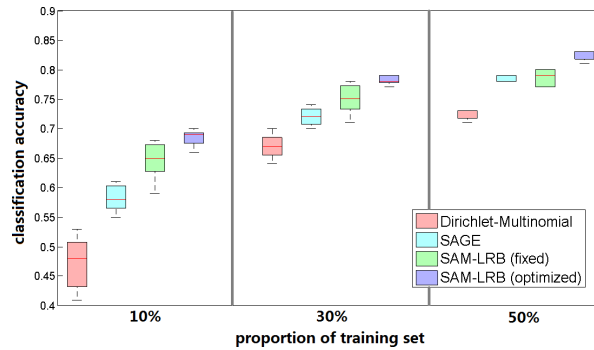

Figure 3: Classification accuracy on `20Newsgroups` data. The proportion of training data varies in $\{10\%, 30\%, 50\%\}$.

One can find that, SAGE outperforms Dirichlet-Multinomial model especially in case of limited training data, which is consistent to the observations in [10]. Moreover, with random and fixed variational variables, the SAM-LRB model performs further better or at least comparably well. If the variational variables are updated to tighten the lower-bound, the performance of SAM-LRB is substantially the best, with a 10%~20% relative improvement over SAGE. Table 1 also reports the average computing time of SAGE and SAM-LRB. We can see that, by avoiding the log-sum-exp calculation, SAM-LRB (fixed) performs more than 7 times faster than SAGE, while SAM-LRB (optimized) pays for updating the variational variables.

Table 1: Comparison on average time costs per iteration (in minutes).

| method | SAGE | SAM-LRB (fixed) | SAM-LRB (optimized) |
|---|---|---|---|
| time cost (minutes) | 3.8 | 0.6 | 3.3 |

## 4.2 Unsupervised Topic Modeling

We now apply our unsupervised SAM-LRB model to the benchmark `NIPS` data[2]. Following the same preprocessing and evaluation as in [10, 26], we have a training set of 1986 documents with 237,691 terms, and a testing set of 498 documents with 57,427 terms.

For consistency, SAM-LRB is still compared with Dirichlet-Multinomial model (variational LDA model with symmetric Dirichlet prior) and SAGE. For all these unsupervised models, the number of latent topics is varied from 10 to 25 and then to 50. After unsupervised training, the performance is evaluated by perplexity, the smaller the better. The performances of 5 independent runs by each method are illustrated in Fig. 4, again in terms of box-plots.

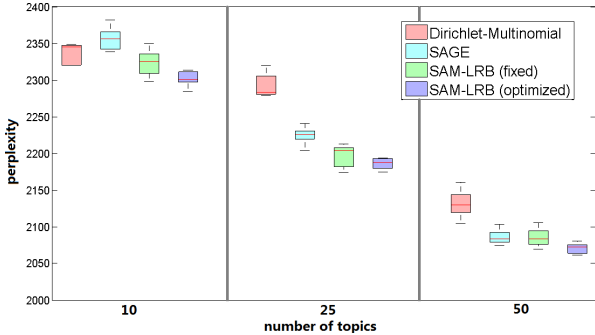

Figure 4: Perplexity results on `NIPS` data.

As shown, SAGE performs worse than LDA when there are few number of topics, perhaps mainly due to its strong equality assumption on background. Whereas, SAM-LRB performs better than both LDA and SAGE in most cases. With one exception happens when the topic number equals 50, SAM-LRB (fixed) performs slightly worse than SAGE, mainly caused by inappropriate fixed values of variational variables. If updated instead, SAM-LRB (optimized) performs promisingly the best.

## 4.3 Multifaceted Modeling

We then proceed to test the multifaceted modeling by SAM-LRB. Same as [10], we choose a publicly-available dataset of political blogs describing the 2008 U.S. presidential election[3] [11]. Out of the total 6 political blogs, three are from the right and three are from left. There are 20,827 documents and a vocabulary size of 8284. Using four blogs for training, our task is to predict the ideological perspective of two unlabeled blogs.

On this task, Ahmed and Xing in [1] used multiview LDA model to achieve accuracy within $65.0\% \sim 69.1\%$ depending on different topic number settings. Also, support vector machine provides a comparable accuracy of $69\%$, while supervised LDA [3] performs undesirably on this task. In [10], SAGE is repeated 5 times for each of multiple topic numbers, and achieves its best median

result 69.6% at $K = 30$. Using SAM-LRB (optimized), the median results out of 5 runs for each topic number are shown in Table 2. Interestingly, SAM-LRB provides a similarly state-of-the-art result, while achieving it at $K = 20$. The different preferences on topic numbers between SAGE and SAM-LRB may mainly come from their different assumptions on background lexical distributions.

Table 2: Classification accuracy on `political blogs` data by SAM-LRB (optimized).

| # topic ($K$) | 10 | 20 | 30 | 40 | 50 |
|---|---|---|---|---|---|
| accuracy (%) median out of 5 runs | 67.3 | **69.8** | 69.1 | 68.3 | 68.1 |

## 5   Concluding Remarks

This paper studies the sparse additive model for document modeling. By employing the double majorization technique, we approximate the log-sum-exponential term involved in data log-likelihood into a quadratic lower-bound. With the help of this lower-bound, we are able to conveniently relax the equality constraint on background log-space distribution of SAGE [10], into a low-rank constraint, leading to our SAM-LRB model. Then, after the constrained optimization is transformed into the form of RPCA's objective function, an algorithm based on accelerated proximal gradient is adopted during learning SAM-LRB. The model specification and learning algorithm are somewhat simple yet effective. Besides the supervised version, extensions of SAM-LRB to unsupervised and multifaceted scenarios are investigated. Experimental results demonstrate the effectiveness and efficiency of SAM-LRB compared with Dirichlet-Multinomial and SAGE.

Several perspectives may deserve investigations in future. First, the accelerated proximal gradient updating needs to compute SVD decompositions, which are probably consuming for very large scale data. In this case, more efficient optimization considering nuclear norm and $\ell_1$-norm are expected, with the semidefinite relaxation technique in [16] being one possible choice. Second, this paper uses a constrained optimization formulation, while Bayesian tackling via adding conjugate priors to complete the generative model similar to [8] is an alternative choice. Moreover, we may also adopt nonconjugate priors and employ nonconjugate variational inference in [27]. Last but not the least, discriminative learning with large margins [18, 30] might be also equipped for robust classification. Since nonzero elements of sparse $S$ in SAM-LRB can be also regarded as selected feature, one may design to include them into the discriminative features, rather than only topical distributions [3]. Additionally, the augmented Lagrangian and alternating direction methods [9, 29] could be also considered as alternatives to the proximal gradient optimization.

## Footnotes

[1]Following [10], we use the training/testing sets from http://people.csail.mit.edu/jrennie/20Newsgroups/

[2]http://www.cs.nyu.edu/~roweis/data.html

[3]http://sailing.cs.cmu.edu/socialmedia/blog2008.html

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
