[Reviews · NeurIPS 2013]

Submitted by Assigned_Reviewer_4

This paper presents a model inspired by the SAGE (Sparse Additive GEnerative) model of Eisenstein et al. The authors use a different approach for modeling the "background" component of the model. SAGE uses the same background model for all; the authors allow different backgrounds for different topics/classification labels/etc., but try to keep the background matrix low rank. To make inference faster when using this low rank constraint, they use a bound on the likelihood function that avoids the log-sum-exp calculations from SAGE. Experimental results are positive for a few different tasks.

Quality:

The paper is solid. The idea of using a low-rank background matrix makes a lot of sense for SAGE, and the use of the double majorization bound and robust PCA learning algorithm seem like good ideas. It works in practice in terms of classification accuracy and perplexity, doing at least as well as SAGE but is simultaneously faster. All in all, I think it's a nice contribution. The paper is clear and the approach is sound. I had some questions, which are listed below, but in general they are minor things.

I would have liked to have seen a little bit more analysis/perspective on why it's working better than SAGE, especially if the reason happens to relate to the use of multiple background vectors with the low-rank constraint. One could also try SAGE with multiple background vectors, right? Presumably this would not work as well, because something like the low-rank constraint would be needed to help bias the learning, but I would prefer more discussion on what exactly the multiple background vectors buys you in practice.

Clarity:

For the most part, the paper is clearly-written. It does draw upon lots of other work which is not fully described for space reasons, but the core ideas are mostly self-contained. I did have some questions here and there, which perhaps the authors can clarify.

I deduce from Algorithm 1 that \nu is set by hand to be 0.05. But is \mu also set by hand? Were the same values used for all experiments or were they tuned for each experiment separately? How sensitive is the performance to the particular values used? Giving some more perspective on this point with more experimental results would be very helpful.

In Figure 2, what is lb-quad-loc? I couldn't find it in the text. I suspect it's lb-quad-j, which is in the text. I guess lb-quad-j should be changed to lb-quad-loc. The figures are very difficult to understand in black-and-white; please make them easier to read, both by increasing the size of the fonts used and using a representation that can be understood in black and white.

Around line 140, "bound proposed by Jaakkola and Jordan [6]" -- I think you meant [14] there instead of [6].

Originality:

This paper is not extremely original, but it's certainly not trivial either. I think that several members of the NIPS community could have come up with something like this if they took the time to think about it, but some points in here are rather neat, like the improvement through optimization of the variational parameters in the bound and the connection to robust PCA.

Significance:

SAGE has been used by a few research groups and seems to be generally known by researchers, so having an improvement to SAGE has the potential for significant impact. I like also that there are some simple ways to extend the approach due to its clean formulation (e.g., different norms could be considered). At the end of the day, it's incremental work, but it's done well and should get published at some point.

Summary: The main ideas are sensible, clearly-described, and lead to improvements in speed and classification accuracy over a standard baseline. I think this is a pretty solid submission.

Submitted by Assigned_Reviewer_5

Sparse additive models represent sets of distributions over large vocabularies as log-linear combinations of a dense, shared background vector and a sparse, distribution-specific vector. The paper presents a modification that allows distributions to have distinct background vectors, but requires that the matrix of background vectors be low-rank. This method leads to better predictive performance in a labeled classification task and in a mixed-membership LDA-like setting.

This is a good paper, but clarity could be improved. Low-rank background distributions seem to provide meaningful improvements in generalization performance. The connection to Robust PCA is a great example of theoretical analysis leading to new algorithms. I felt some important details were missing, like some outline of the Robust PCA algorithm. What is a double majorization bound?

The paper is also missing the big picture. What does it mean to have a low-rank background distribution? Is it learning connections between distributions, that might be interpretable? It's not exactly intuitive. Shouldn't any variation from the mean distribution be accounted for in the distribution-specific vectors? Is this a way to get around overly harsh L1 regularization?

Be more clear about why Eq 3 is saving time. The original log sum exp has M exp evaluations and a log, while in Eq 3, the f() function (is there really not a more interesting name?) has two exps and a divide for each i. Is the advantage in calculating the derivative?

If the point of Fig. 2b,2c is that the variables converge, then say that in the caption. I also don't get the significance of the checkerboard pattern in Fig 1.

Are there parentheses missing in Eq. 7? It doesn't matter mathematically, but line 194 on p4 makes it sound like mu and nu are distinct weights.

I'd like to see a bit more about the evaluation procedure for Fig 4. Are these numbers comparable? It looks like the four models are just different ways of producing topic distributions, so it should be possible to feed an identical data structure to the same model-agnostic perplexity-calculation code. Is that what happened?

In 4.3, it's not clear to me what the task is: are we estimating the polarity of the two blogs as a whole, or the individual posts in the blogs?

The fact that different models peak at 20 and 30 makes me wonder how different these results really are. Is 69.8 vs 69.1 significant, in any of its many senses?

Many grammatical errors.
Summary: An interesting extension to an existing model, with a clever connection to not-obviously-related work that results in a good algorithm.

Submitted by Assigned_Reviewer_6

Summary: Previous work on SAGE introduced a new model for text. It built a lexical distribution by adding deviation components to a fixed background. The model presented in this paper SAM-LRB, builds on SAGE and claims to improve it by two additions. First, providing a unique background for each class/topic. Second, providing an approximation of log-likelihood so as to provide a faster learning and inference algorithm in comparison to SAGE.

Strengths:
a) A faster inference and learning algorithm by estimation of log-likelihood
b) A more general model than SAGE, allowing different background components for each class/topic

Weaknesses:
a) In SAGE, the key idea behind a single background component across all topics/classes was to have a drop-in replacement for a stop word list in models like LDA. Having a different background across each class/topic makes the assumption that all topics/classes have no common words, and makes model more complex, without a clear benefit.
b) Missing a related work section. It would be nice to see this model compared with related work such as 'Shared Components Topic Models, Gormley, Dredze, Durme, Eisner et.al"
c) The experiments demonstrate that SAM-LRB works well when the training data is limited. It would be nice to see when all training data is used, as shown in the SAGE paper.
d) The experiments do not show significant speed improvements when using an optimized version on SAM-LRB. It would be nice to see experiments where a larger vocabulary size is used (so that sum-of-exp becomes significant) and the results compared to SAGE and traditional LDA models.

Quality: This paper seems technically sound. The authors do give experiment results supporting their claims. More experiments as suggested above in weaknesses would make a stronger case.

Clarity: The paper is written clearly. Although the first term in equation 1, is a bit confusing, as it incorrectly demonstrates independence between document class and the background and additive components. Also, while making case for introducing different background components, it would be nice to see how it addresses the redundancy issue mentioned in SAGE.

Originality: This paper is a mildly novel combination of ideas from SAGE and using an approximation on log-likelihood. More significant is its reduction of computation time for learning and inference.

Significance: This model seems useful for the case of limited training data, and also speed when compared to SAGE and LDA, although further experiments would help make this case stronger.
Summary: This paper is an extension of SAGE and the most significant contribution is a faster inference and learning algorithm. Some comparisons with related work, and adding more experiments would make a stronger case.
Author Feedback

Author rebuttal: To Assigned_Reviewer_4:
1. Concern: Expect more analysis on why SAM-LRB outperforms SAGE.
Reply: In SAGE, the exactly-the-same constraint on the background is sometimes too strong. When applied to the low-rank background cases illustrated by Fig.1, it cannot correctly decouple the back/foreground addition. An example on text modeling will be considered to add if the space limit allows. Thanks for your suggestion.

2. Concern: The setting and sensitiveness of \mu and \nu.
Reply: Once the derivation coincides with robust PCA, we call the accelerated proximal gradient algorithm in [19]. Therein, parameter \mu is adaptively determined; and parameter \nu, trading-off between low-rank and sparsity, is suggested to be around 0.05~0.1 and able to provide robust performance in experiments.

3. Concern: The legends, fonts and colors in Fig.2.
Reply: The legend lb-quad-loc refers to lb-quad-j in the text, the latter of which is a typo. Thanks for the elaborative correction. The fonts and colors will be revised for easier understanding.

4. Concern: The citation number around line 140.
Reply: The citation number is a typo. Thank you very much for pointing it out.

To Assigned_Reviewer_5:
1. Concerns: More details on, e.g. robust PCA, double majorization bound.
Reply: More descriptions on these details will be considered to be added if the space allows.

2. Concerns: More intuition on low-rank background distributions.
Reply: In SAGE, the exactly-the-same constraint on background is sometimes too strong. When applied to the low-rank background like Fig.1, it cannot correctly decouple the back/foreground addition. Thanks for your suggestion.

3. Concerns: More explanations on Eq.(3).
Reply: The key reason of time saving by Eq.(3) is the quadratic form over x. All remaining values involved only \xi are auxiliary, and can be calculated/updated in a lazy manner, i.e., calculate once and update seldom.

4. Concerns: Caption of Fig 2.
Reply: Yes, Fig.b&c illustrate the fast convergence of function and variables, resp. The caption will be revised accordingly.

5. Concerns: Description of Eq.(7).
Reply: Parameters \mu and \nu are distinct values, same as in the formulation of robust PCA. In Algorithm 1 and in the accelerated proximal gradient algorithm of [19], \mu is determined adaptively and \nu is set around 0.05~0.1, providing robust performances in experiments.

6. Concerns: Evaluation for Fig.4.
Reply: The perplexity is calculated in the same standard form. It can be observed that, as the topic number decreases, the advantages of the low-rank background in SAM-LRB become more and more obvious.

7. Concerns: Performance of Sec 4.3.
Reply: We are to predict individual posts in the two blogs. For this difficult task, current state-of-the-art performance is 69.6 by SAGE at K=30, and SAM-LRB reaches 69.8 at K=20. Thank you for valuable suggestions.

To Assigned_Reviewer_6:
1. Concerns: Explanation of benefit of SAM-LRB.
Reply: In SAGE, the exactly-the-same constraint on background is sometimes too strong. When applied to the low-rank background cases illustrated by Fig.1, it cannot correctly decouple the back/foreground addition.

2. Concerns: More related work discussions.
Reply: Thanks for your valuable suggestion. More discussions will be added accordingly.

3. Concerns: The size of training data in Fig.3.
Reply: Results of using increasingly more training data will be reported accordingly, which also show promising performance of SAM-LRB.

4. Concerns: Expect results on a larger vocabulary size.
Reply: Experimental results on data with larger vocabulary sizes will be reported accordingly. Therein, as you pointed out, the sum-of-exp becomes more significant, and the computational advantages of SAM-LRB(fixed) over SAGE also becomes more significant. Your suggestion would help improve the paper's quality, thanks a lot.